# Effects of Repeated Forward Versus Repeated Backward Sprint Training on Physical Fitness Measures in Youth Male Basketball Players

**DOI:** 10.3390/sports14010016

**Published:** 2026-01-04

**Authors:** Ghofrane Arbi, Yassine Negra, Aaron Uthoff, Senda Sammoud, Patrick Müller, Helmi Chaabene, Younes Hachana

**Affiliations:** 1Research Laboratory (LR23JS01) “Sport Performance, Health & Society”, Higher Institute of Sport and Physical Education of Ksar Saïd, University of “La Manouba”, Manouba 2037, Tunisia; ghofranearbi1998@gmail.com (G.A.); yassinenegra@hotmail.fr (Y.N.); senda.sammoud@gmail.com (S.S.); hachanayounes@gmail.com (Y.H.); 2Sports Performance Research Institute New Zealand (SPRINZ), AUT Millennium, School of Sport and Recreation, Auckland University of Technology, Auckland 0632, New Zealand; aaron.uthoff@aut.ac.nz; 3Institut Supérieur de Sport et de l’Education Physique du Kef, Université de Jendouba, Le Kef 7100, Tunisia; 4Department of Cardiology and Angiology, University Hospital Magdeburg, 39120 Magdeburg, Germany; patrick.mueller@med.ovgu.de

**Keywords:** adolescents, team sport, athletic performance, sport-specific training

## Abstract

This study aimed to examine the effects of eight weeks of repeated backward sprint training (RBST) versus repeated forward sprint training (RFST) on physical fitness measures. Twenty-three postpubertal male basketball players (mean age = 15 years) were randomly assigned to either an RBST group (*n* = 12) or an RFST group (*n* = 11). Pre- and post-intervention assessments measured 5-, 10-, and 20 m sprint performance, Y-agility, 505 change of direction (CoD) speed, countermovement jump (CMJ), standing long jump (SLJ), and RSA (RSA best time [RSA_best_] and RSA mean time [RSA_mean_]). The RBST group significantly improved in all measures (*p* < 0.05; 6.11 to 19.25%; Effect size [ES] = 0.32 to 1.05) except RSA_best_. The RFST group significantly improved 10 m sprint, SLJ, RSA_best_, and RSA_mean_ (*p* < 0.05; 6.25 to 17.84%; ES = 0.05 to 0.80). Between-group analysis revealed that RBST outperformed RFST in Y-agility and SLJ (*p* < 0.05; ES = −1.03 and 0.16, respectively). RBST was more effective for improving agility and lower-body power, while RFST provided a slight advantage in peak RSA. These findings suggest that incorporating task-specific sprint training may optimize physical performance in male youth basketball players.

## 1. Introduction

Athletic capacities, including sprint speed, jumping ability, change-of-direction (CoD) performance, agility, and repeated-sprint ability (RSA), play a crucial role in determining overall basketball performance [1,2,3,4]. These athletic qualities significantly influence a player’s effectiveness on the court [5,6]. Research has shown that elite basketball players typically demonstrate higher levels of muscular strength, speed, and overall movement proficiency, such as superior acceleration, sprinting, jumping, CoD, and RSA, when compared with sub-elite players [7,8].

Basketball is characterized by its highly intermittent nature, with players performing approximately 1050 distinct movements during a single competitive match, equating to a change in speed or direction roughly every two seconds [9]. During gameplay, elite basketball athletes typically cover about 991 m through high-intensity efforts, executing on average 40–60 maximal jumps and 50–60 rapid accelerations or CoD [9]. Moreover, it has been reported that nearly half (≈52%) of all sprints in basketball involve at least one fast CoD, emphasizing that the ability to efficiently execute directional changes, both with and without ball possession, is a fundamental determinant of performance [10]. Although these athletic attributes tend to improve naturally in young athletes as a result of growth and maturation processes [11], targeted training programs can further accelerate and enhance their development [12]. With the increasing competitiveness of youth basketball, optimizing training interventions that enhance sprinting, the physical and cognitive components of agility, jumping ability, and RSA are essential for success in the sport.

Given the sport’s multifaceted physical demands, diverse training methods, ranging from non-specific to sport-specific interventions, are commonly employed to improve strength, speed, power, and RSA in youth basketball players [13,14,15,16,17]. The principle of specificity asserts that training adaptations are specific to the imposed stimuli, with optimal performance improvements occurring when the training stimulus closely matches the neuromuscular and physiological demands of the target movement [18]. Research supports this concept, showing that training methods with strong biomechanical and physiological similarities to a given task, facilitated by shared locomotive neural pathways [19,20], result in greater transfer to sport-specific movements. In basketball players, such targeted training has been linked to enhanced sprinting speed, CoD ability, jumping performance, and RSA [13,19,21,22,23]. Although specific training methods are generally understood to directly resemble the demands of an athletic movement [18,24], growing evidence suggests that practicing movements in the opposite direction can induce adaptations that enhance forward-directed performance [25]. Specifically, backward running (BR) has been proposed as an effective training strategy for improving forward sprinting [26], jumping, CoD speed, and RSA [25,26,27,28,29], as both movement patterns are thought to be governed by the same central pattern generators responsible for locomotion [19].

Additionally, BR training has been shown to enhance cardiovascular fitness, as it requires roughly 28–35% more energy than forward running (FR) at comparable velocities due to a decreased utilization of elastic energy [30,31] and greater reliance on isometric and concentric muscular actions [32], making it a particularly effective stimulus for improving athletic qualities dependent on cardiorespiratory fitness, such as RSA [9]. Recent evidence suggests that performing repeated BR training results in positive adaptations to not only athletic movements in line with cardiovascular fitness but also explosive movements [27,28]. For example, Negra et al. [28] and Bouguezzi et al. [27] provide strong evidence for the effectiveness of repeated BR training in youth soccer players, demonstrating that integrating BR training into a sports program for six to eight weeks leads to significant improvements in sprinting, CoD speed, jumping, and RSA beyond those achieved through standard soccer training alone. However, while the growing body of research into repeated BR training is promising compared to traditional sports training, no studies have compared the effects of repeated BR versus repeated FR.

The existing literature suggests that task-specific training can effectively improve basketball-related physical performance capabilities, with repeated BR potentially offering superior adaptations [26,28,29]. However, with only two studies currently available [27,28], both of which were in youth soccer players, and neither comparing the effects of repeated BR to repeated FR, there is a gap in our understanding of which direction of training leads to the most remarkable physical fitness adaptations in youth basketball players. To address this gap, the present study is the first to directly compare the effects of repeated BR training and repeated FR training on sprinting, CoD speed, agility, jumping ability, and RSA in youth basketball players. We hypothesized that both repeated FR training and repeated BR training would enhance physical performance, but that repeated BR training would elicit greater improvements [26,33].

## 2. Materials and Methods

### 2.1. Participants

Following Negra et al. [33], an a priori power analysis was conducted with an alpha significance level (α) of 0.05 and a desired statistical power of 95%. The results indicated that 18 participants would be adequate to detect a meaningful interaction effect (effect size Cohen’s *d* = 0.94 for the RSA total time parameter). To accommodate possible dropouts, 23 postpubertal male basketball players were ultimately recruited. Participants were matched according to maturation status and randomly assigned to either the RFST group (*n* = 11) or the RBST group (*n* = 12). All participants had an average of 9.0 ± 1.1 years of structured basketball training and reported being in good health, with no musculoskeletal or tendon injuries in the previous six months. Table 1 presents the anthropometric characteristics of both groups. Biological maturity was assessed using the maturity offset method [34], and the following prediction equation was applied:

Maturity offset = −7.999994 + (0.0036124 × age × height). Based on the predicted maturity offset, athletes were classified according to their proximity to peak height velocity (PHV). Specifically, participants were categorized into three maturity groups: pre-PHV (−3 to −1 years from PHV), circa-PHV (−1 to +1 years from PHV), and post-PHV (+1 to +3 years from PHV). All participants demonstrated positive maturity offset values, confirming that the vast majority were postpubertal. Specifically, only two players in the RBST group and two players in the RFST group were classified as pubertal. Prior to randomization, players were matched according to their maturity offset to ensure similar biological maturation between groups. Before randomization, participants were matched based on their biological maturation status. All postpubertal players were randomly distributed evenly across the two groups, and the four pubertal players were also balanced, with two randomly assigned to the RFST group and two to the RBST group. The mean maturity offset values did not differ significantly between the RFST and RBST groups at baseline (RFST: 1.86 ± 1.01 years; RBST: 1.74 ± 0.68 years; *p* = 0.78, 95% CI = −0.28 to 0.22), indicating balanced maturation status distribution between the groups

### 2.2. Experimental Approach to the Problem

We examined and compared the effects of 8 weeks of biweekly in-season repeated FR sprint training (RFST) versus repeated BR sprint training (RBST) on various measures of physical fitness in youth basketball players. A regional team of male youth basketball players was matched for maturation status and randomly assigned to an RFST group and a BRST group. Two familiarization sessions were conducted over two weeks before baseline testing to ensure participants were accustomed to the testing procedures. Pre- and post-intervention testing assessed changes in physical performance, including linear sprint speed (5-, 10-, and 20 m sprints), 505 CoD speed, agility (Y-agility test), jump height (countermovement jump [CMJ]), jump distance (SLJ), and repeated sprint ability (RSA). All tests were scheduled at least 48 h after the last training session or match and were carried out at the same time of day (18:00–19:30). Testing occurred over three days with anthropometric measurements, linear sprint speed, CoD speed, and agility testing conducted on the first day, jump testing on the second day, and RSA on the third day. All participants received the treatment as allocated. No training or test-related injuries were reported.

All experimental procedures and any potential risks were thoroughly explained. Written informed consent was obtained from the participants’ parents or legal guardians, and assent was secured from the participants themselves. The study protocol received approval from the local Institutional Review Committee of the Higher Institute of Sports and Physical Education of Ksar Said (Approval No. LR, R: 203–207) and was carried out in accordance with the most recent version of the Declaration of Helsinki [19].

### 2.3. Linear Sprint Speed Time

Twenty-meter sprint performance was assessed at 5, 10, and 20 m intervals using a single-beam electronic timing system (Wittygate, Microgate, SRL, Bolzano, Italy). Participants started from a standing split-stance position, placing their lead foot roughly 0.3 m behind the initial infrared timing gate, which was positioned 0.75 m above the ground to register trunk movement and reduce the chance of false signals from limb motion. Four single-beam photoelectric gates were set up along the sprint pathway. Participants were instructed to avoid rocking or taking any preparatory steps before starting. A rest period of three minutes was provided between trials, and the best time from the two attempts was used for subsequent analysis.

### 2.4. Y-Agility Test

The Y-agility test was administered following the protocol described by Lockie et al. [35]. The Witty light-based timing system (Microgate, SRL, Bolzano, Italy) was used to record completion time and control the reactive stimulus. Timing gates were positioned at a width of 1.5 m and a height of 1.2 m. Participants began 0.3 m behind the start line and sprinted maximally for 5 m in a straight line. Upon reaching the decision point, they executed a CoD at a 45° angle to either the left or right, followed by a 5 m sprint through the finish gates. A reactive green arrow stimulus was used to indicate the required CoD speed’s direction, appearing approximately 40–45 ms after passing the starting gate. Each participant completed two trials, with the fastest time used for analysis. A 90 s rest period was provided between trials.

### 2.5. Countermovement Jump Test

During the CMJ, participants started from an upright stance and executed a quick countermovement by bending at the hips and knees, followed by an explosive upward extension to achieve a maximal vertical jump. To limit the influence of arm swing, they were instructed to keep their hands placed on their hips throughout the movement. Jump height was measured using an optoelectronic system (Optojump Next, Microgate, SRL, Bolzano, Italy). Each participant performed three attempts, and the greatest jump height was used for analysis. A recovery period of 90 s was provided between efforts.

### 2.6. Standing Long Jump Test

Participants stood just behind the starting line with their feet shoulder-width apart and their arms relaxed at their sides. Upon hearing the command “ready, set, go,” they performed a quick countermovement before explosively jumping forward as far as possible. They were required to land on both feet simultaneously and maintain balance, without falling forward, backward, or to the side. Jump distance was measured from the starting line to the heel of the foot closest to the take-off point using a tape measure, accurate to the nearest centimeter. Each participant completed three attempts, with the longest jump retained for analysis, and a 90-s rest interval was provided between trials.

### 2.7. Repeated Sprint Ability Test

The RSA test was administered using the same photocell timing system employed for the linear sprint and 505 change-of-direction assessments (Wittygate, Microgate, SRL, Bolzano, Italy). After completing a standardized warm-up, participants performed a single familiarization shuttle sprint (15 + 15 m with a 180° turn) to establish a reference time for the subsequent RSA test [36]. Participants then rested for five minutes before beginning the protocol. During the first sprint, they were required to reach at least 97.5% of their reference time; if this criterion was not achieved, they rested for an additional five minutes before repeating the attempt [36]. This procedure was implemented to prevent participants from adopting a pacing strategy. All individuals successfully met the criterion on their first sprint. The RSA test consisted of six 15 m shuttle runs incorporating a 180° change of direction, each separated by 14 s of passive recovery [36]. Three seconds prior to each sprint, participants assumed a ready split-stance position with their lead foot 0.3 m behind the start line and maintained this posture until the start cue. They then sprinted 15 m at maximal speed, contacted the turning line with one foot, executed a 180° turn, and returned to the start line as fast as possible. Participants were instructed to perform every sprint with maximal effort. The fastest sprint time (RSAbest) and the mean sprint time (RSAmean) were recorded for analysis.

### 2.8. The Training Program

The RFST and RBST sessions were integrated into the regular basketball training routine of the intervention groups following a standard warm-up. These sessions replaced 10 to 15 min of low-intensity basketball drills on Tuesdays and Thursdays over 8 weeks (Table 2). Both groups completed an equal total running distance per session. The BR technique was reinforced using coaching cues adapted from Uthoff et al. [26] and Negra et al. [33], including “slight forward lean of the chest”, “use arm mechanics similar to forward running”, and “high heel recovery of the swing leg”. Players were instructed to sprint with maximal effort for all repetitions, completing the prescribed 20 m running distance as fast as possible. Each session consisted of two to four sets, with seven repetitions per set. Rest intervals were 20 s between repetitions and 4 min between sets. Following the RFST and RBST sessions, players continued their regular basketball training. Each training session lasted approximately 80–90 min in total, consisting of a 15–20 min standardized warm-up (mobility, dynamic stretching, and activation drills), 10–15 min of the RFST or RBST intervention, and approximately 40–45 min of technical–tactical basketball drills and scrimmage play. Following the RFST and RBST sessions, players continued their regular basketball training, which included offensive and defensive drills as well as small-sided games. At the end of each session, a 5–10 min cool-down with low-intensity jogging and stretching was performed.

### 2.9. Statistical Analyses

Data are presented as means and standard deviations (SDs). Normal distribution was assessed using the Shapiro–Wilk test. A 2 (group: RBST and RFST) × 2 (time: pre and post) repeated-measures ANOVA was conducted to assess intervention effects on the dependent variables. If significant interactions were detected (i.e., significant F value), group-specific post hoc tests (i.e., paired *t*-tests) were used. The alpha level of significance was set at *p* < 0.05. To quantify the magnitude of performance changes within and between groups, percentage change and Cohen’s *d* effect size (ES) statistics were calculated [37]. ES magnitudes were classified as trivial (<0.2), small (≥0.2 to 0.49), moderate (≥0.5 to 0.79), and large (>0.8) [37]. To evaluate the practical relevance of performance changes, the smallest worthwhile change (SWC) was determined using the pooled standard deviation (SD) of pre-training session scores across all groups. The SWC was converted to a percentage for each performance variable, with thresholds set at small (SWC = 0.2 × SD), moderate (MWC = 0.6 × SD), and large (LWC = 1.2 × SD) [26,27,28,38]. All data analyses were performed using SPSS 25.0 (IBM Corp., Armonk, NY, USA).

## 3. Results

Within-group changes from pre- to post-training and between-group comparisons for all physical fitness measures are displayed in Table 3. At baseline, no significant between-group differences were observed with respect to anthropometric characteristics and maturity offset. Based on the maturity offset values (Table 1), the majority of participants were classified as postpubertal; however, four players (two in each group) fell within the pubertal range. Accordingly, maturation status was not uniform across all participants, but the distribution of pubertal and postpubertal athletes was balanced between groups. Likewise, no between-group differences were recorded at baseline for any measure of physical fitness except for the 5 m, CMJ, SJ, and RSA tests.

### 3.1. Linear Sprint

The statistical analysis showed a significant effect for time (*p* = 0.000) and a non-significant group × time interaction for the 5, 10, and 20 m sprint tests (*p* = 0.803, 0.923, and 0.749, respectively, for the 5, 10, and 20 m sprint tests). Significant pre-to-post within-group changes were found for both the RBST group (*p* < 0.05; 0.98 to 5.32%; ES = 0.91 to 1.05), and corresponding 95% confidence intervals (CIs) of 1.01 to 1.09 (5 m), 0.93 to 1.02 (10 m), and 0.81 to 0.99 (20 m), and the RFST group (*p* < 0.05; −3.24 to −4.58%; ES = 0.50 to 0.80). The 95% CI were −0.76 to 0.84 (5 m), 0.57 to 0.71 (10 m), and 0.37 to 0.64 (20 m). Our analysis revealed that 83% of the RBST group (*n* = 10) improved 5 m, 10 m, and 20 m sprint performance beyond the SWC. It was observed that 100% of the RFST group (*n* = 11) improved 5 m sprint performance beyond the SWC, with 64% (*n* = 7) improving 10 m sprint performance, and 45% (*n* = 5) showing improvement in 20 m sprint performance beyond this threshold (Figure 1). No between-group differences were observed (*p* > 0.05).

### 3.2. 505 Change of Direction

The statistical analysis showed a significant effect for time (*p* = 0.009) and a non-significant group × time interaction (*p* = 0.188) for 505 CoD performance. Moderate, significant pre-to-post within-group improvements were found for the RBST group (*p* < 0.05, 6.48% ES = 0.62, [95% CI: 0.43 to 0.77]), while trivial, non-significant improvements were observed in the RFST group (*p* > 0.05; 2.18%, ES = 0.18, [95% CI: −0.04 to 0.38]). Our analysis revealed that 58% (*n* = 7) of the RBST group and 45% (*n* = 5) of the RFST group improved 505 CoD performance above the SWC (Figure 1). While RBST was found to have a small beneficial effect on 505 CoD performance compared to RFST, no significant between-group differences were observed (*p* > 0.05).

### 3.3. Y-Agility

The finding showed a significant effect for time and group × time interaction (*p* = 0.000) for the Y-agility test. Post hoc analyses demonstrated a large within-group performance improvement from pre- to post-testing for the BRST group (*p* < 0.01; 13.27%; ES = 1.33, [95% CI: 1.20 to 1.43]), while the RFST group showed a trivial, non-significant pre-post change in performance (*p* > 0.05; 0.68%; ES = 0.05, [95% CI: −0.09 to 0.18]). In terms of individual response rates, our results revealed that all but one athlete (92%) in the RBST group improved their Y-agility test performance more than the SWC, and 50% of the group (*n* = 6) improved more than the MWC. Alternatively, only a single athlete (9%) in the RFST group improved Y-agility performance more than the SWC (Figure 1). Large, significant between-group differences were observed, with the RBST group improving more than the RFST group (*p* < 0.01; *d* = 1.03).

### 3.4. Countermovement Jump

Findings showed a significant effect for time (*p* < 0.01) and a non-significant group × time interaction (*p* = 0.798) for CMJ performance. Within-group analysis from pre- to post-test showed small significant improvements in both the RBST (*p* < 0.05; 6.11%; ES = 0.38 [95% CI: 0.38 to 0.41) and RFST (*p* < 0.05; 6.11%; ES = 0.38 [95% CI: 0.34 to 0.41) groups (*p* < 0.05; 6.25%; ES = 0.34 [95% CI: −3.19 to 3.18]). Regarding individual responses, 83% and 33% of the RBST group (*n* = 10 and 4) and 73% and 25% (*n* = 8 and 2) of the RFST improved CMJ performance above the SWC and MWC, respectively (Figure 1). No between-group differences were observed (*p* > 0.05).

### 3.5. Standing Long Jump

Results showed a significant effect for time (*p* < 0.01) and group × time interaction (*p* = 0.012) for the SLJ. Within-group analysis revealed that both the RBST and RFST groups improved performance (*p* < 0.05; RBST: 2.69%; ES = 0.30 [95% CI: 0.19 to 0.40]; RFST: 1.34%; ES = 0.15 [95% CI: 0.02 to 0.27]). Additionally, 67% (*n* = 8) of the RBST group improved their performance above the SWC, while only a single athlete (9%) improved above this threshold (Figure 1). Trivial, significant between-group differences were observed, with the RBST group improving more than the RFST group (*p* < 0.05; *d* = 0.16).

### 3.6. Repeated Sprint Ability

The statistical analysis showed a significant effect for time (*p* < 0.01 for the RSA_best_ and RSA_mean_, respectively) and no significant group × time interaction (*p* = 0.358 and 0.593 for the RSA_best_ and RSA_mean_, respectively) for all the RSA outcomes. Within-group analysis revealed that RFST led to small, significant improvements from pre-to post-test for both the RSA_best_ and RSA_mean_ (*p* < 0.05, 2.58 to 2.91%). Effect sizes indicated small changes for RSA_best_ (ES = 0.45, 95% CI: 0.12 to 0.76) and RSA_mean_ (ES = 0.43; 95% CI: 0.10 to 0.74), while the BRST led to small-to-moderate improvements to both RSA outcomes, with only RS_mean_ reaching statistical significance (*p* < 0.05; 2.16%, ES = 0.62; [95% CI: 0.43 to 0.77). Regarding individual changes, 64% (*n* = 7) of the RFST group and 67% (*n* = 8) of the RBST group improved RSA_best_ performance more than the SWC. Meanwhile, 73% (*n* = 8) of the RFST and 67% (*n* = 8) of the RBST group improved RSA_mean_ performance more than the SWC (Figure 1). While no significant between-group differences were observed for either RSA measure (*p* > 0.05), RFST had a small beneficial effect compared to RBST on RSA_best_.

## 4. Discussion

To the best of the authors’ knowledge, the current study is the first to compare the effects of RBST versus RFST on various measures of physical fitness in youth male youth basketball players. The main findings indicated that both RBST and RFST led to significant within-group improvements across multiple performance measures, including sprinting, CoD speed, jumping, and RSA. However, between-group comparisons revealed that RBST elicited superior adaptations in Y-agility and SLJ performance, while RFST showed a small advantage in RSA_best_. These results suggest that task-specific RBST may provide greater benefits for agility and proxies of muscle power, whereas RFST may be slightly more advantageous for peak repeated sprint performance. These findings are generally consistent with our initial hypothesis, which proposed that both repeated forward and backward sprint training would enhance physical performance, with greater improvements expected following backward sprint training.

Linear sprint speed over short distances is a crucial physical fitness attribute for basketball players, as it directly impacts their ability to accelerate quickly, change direction, and perform fast breaks or defensive movements on the court [1,2,3,4,5,6]. In the present study, both the RBST and RFST groups demonstrated small-to-moderate within-group improvements across the 5, 10, and 20 m sprint distances (ES = −0.50 to −0.89). However, only the RBST group showed statistically significant improvements across all sprint distances, whereas in the RFST group, only the 10 m sprint improvement reached statistical significance, with the 5 m and 20 m changes not meeting this threshold. Regarding individual outcomes, 83% of participants in the RBST group (*n* = 10) demonstrated improvements in 5 m, 10 m, and 20 m sprint times exceeding the SWC_0.2_. In the RFST group (*n* = 11), all participants (100%) showed improvements in 5 m sprint performance beyond the SWC_0.2_, with 64% (*n* = 7) improving 10 m sprint times and 45% (*n* = 5) improving 20 m sprint times beyond this threshold. These findings are broadly consistent with previous literature showing that repeated sprint training, regardless of direction, can improve short-distance sprint performance in youth athletes. For example, Negra et al. [28], reported moderate-to-large within-group improvements over 5, 10, and 20 m distances following an 8-week RBST programme in 13–14-year-old male soccer players (5 m: −9.99%, ES = −1.79; 10 m: −2.39%, ES = −0.74; 20 m: −2.24%, ES = −0.75), with between-group ES of −2.23, −1.24, and −0.70, respectively, when compared to a control group. Supporting this, Bouguezzi et al. [27] found that a 2-day weekly RBST programme over six weeks led to large improvements in 10 m (Δ4.74%, ES = 0.96) and 20 m (Δ3.34%, ES = 1.00) sprint performance in youth male soccer players aged 16 years. Taken together, this evidence suggests that while RFST can yield performance benefits to short linear sprint performance, RBST, particularly when applied biweekly, may induce greater and more consistent improvements in short sprint performance among youth team sport athletes, including basketball players.

In the present study, RBST led to meaningful improvements in both CoD and agility performance, with a higher proportion of athletes in the RBST (58 to 92%) group surpassing the SWC threshold in both domains compared to the RFST (9 to 45%) group. These findings are consistent with previous studies showing that RBST enhances CoD performance in youth athletes (ES = 0.78–1.12) [27,29,33], often producing larger effects than RFST interventions, which typically result in moderate gains (ES = 0.62), even when incorporating direction changes [39]. The notably greater effect of RBST on agility compared to CoD suggests that this training method may target not only the physical components of multidirectional speed but also the cognitive–perceptual elements that underpin reactive agility. To our knowledge, this is the first study to show that RBST can enhance performance in an agility task that incorporates cognitive-perceptual demands, indicating that RBST may provide broader sport-specific adaptations than RFST. However, further studies are needed to substantiate the current findings. The superior effects of RBST on agility performance may be attributed to its influence on both kinetic (e.g., horizontal force and impulse) and kinematic factors (e.g., step length and step frequency), as well as improvements in CoD technique [40]. Backward running requires heightened coordination and proprioceptive control [41], which may foster the development of more effective movement strategies, allowing athletes to better position their bodies for directional changes [42]. Furthermore, the distinct physiological responses to backward running, such as reliance on isometric and concentric muscular actions [30,31] and higher levels of lower limb muscle activation compared to forward running [43,44], are considered to be critical contributors to CoD performance [18]. Unlike forward running, backward running limits visual guidance, requiring athletes to process alternative sensory cues to maintain spatial awareness [45]. This may have important implications for agility tasks that rely on visual tracking of teammates or opponents while moving through space on a court or field. The use of backward running and its derivatives, like backpedaling, has already been embedded in many sports for reactive agility movements and has been proposed as a strategy to increase movement variability, reduce injury risk, and enhance transfer to sport-specific performance [25,46]. Collectively, these attributes could explain why RBST induced more robust improvements in both CoD and agility compared to RFST, offering a more comprehensive training stimulus that encompasses both the neuromechanical and perceptual demands of multidirectional athletic actions. Nevertheless, future studies evaluating the potential neuromechanical factors driving the greater improvements in CoD speed and agility performance after RBST compared to RFST are necessary to deepen our understanding.

In the present study, both RBST and RFST produced small improvements in CMJ-height, though only the RBST group achieved a statistically significant change. Despite similar effect sizes (RBST:ES = 0.34; RFST:ES = 0.40), the proportion of athletes exceeding the SWC was comparable (RBST: 66%; RFST: 72%). In contrast, SLJ performance showed clearer differentiation, with both groups improving significantly, but only the RBST group demonstrating a meaningful effect compared to the RFST group (small vs. trivial), and a markedly higher proportion of athletes surpassing the SWC (67% vs. 9%, respectively). Although the between-group difference was statistically significant, the effect size was trivial (ES = 0.16). These findings align with previous studies showing that RBST can effectively enhance both vertical and horizontal jump performance in youth athletes [27,28]. Additionally, backward running interventions using longer inter-repetition rest periods have also been shown to improve jump performance. Uthoff et al. [26] found moderate gains in CMJ-height (ES = 0.83; Δ9.9%) following an eight-week BR program in adolescent male athletes. Likewise, Sammoud et al. [29] reported a small but significant improvement in SLJ performance (ES = 0.46; Δ5.57%) in young female handball players. The effectiveness of both repeated and non-repeated backward running protocols may stem from the unique neuromuscular demands of backward locomotion, which utilize a less efficient stretch-shortening cycle [30,31] and likely promote strength adaptations transferable to jumping tasks that rely heavily on concentric quadriceps output [47]. While interesting to suggest, the veracity of this theory still requires further investigation to fully support a causal link between lower body muscle actions and adaptations to jumping ability following backward running training.

In the present study, RFST led to small but statistically significant improvements in both RSA_best_ and RSA_mean_ (−2.58% to −2.91%, *p* < 0.05). The RBST group also demonstrated small-to-moderate improvements in both outcomes, although only RSA_mean_ reached statistical significance (−2.16%, *p* < 0.05). These findings align with prior research demonstrating that both RBST and RFST can enhance RSA in youth athletes. RBST [27,28], in particular, has been shown to elicit moderate-to-large improvements in RSA_best_ and RSA_total_ following 8-week biweekly interventions, with consistent benefits reported across multiple studies [27,28]. Similarly, RFST interventions have produced moderate-to-large improvements in RSA_best_ (1.7%; ES = 0.65) and RSA_mean_ (ES = 2.9%; ES = 0.69). Furthermore, earlier RFST intervention studies showed moderate-to-large improvements in RSA outcomes, with effect sizes ranging from 0.65 to 0.69 and percentage changes between 1.7% and 2.9%, across programmes lasting 6 to 9 weeks, and across various populations, including amateur and elite youth athletes, with or without CoD components [33,39]. The improvements observed in the current study are therefore consistent with the magnitude of change reported in the literature, reinforcing the effectiveness of both training modalities for developing RSA outcomes in youth athletes. It is worth noting that, at the same speed, backward sprints generally require greater energy expenditure than forward sprints due to the higher muscle activation and coordination demands involved [25,26]. Additionally, although participants were instructed to sprint as fast as possible in both directions, it is plausible that forward running resulted in higher speeds than backward running. This may have led to greater metabolic demands during forward sprints, which in turn could have more favorably impacted RSA performance. Moreover, while backward sprinting engages more muscle groups and may offer greater neuromuscular stimulation, it generally involves a less efficient movement pattern compared to forward sprinting [25]. This inefficiency can lead to faster fatigue accumulation, which may hinder performance in repeated sprint bouts. In contrast, forward sprints utilize a more familiar and efficient movement pattern, allowing athletes to maintain a higher velocity during repeated sprints [47]. Even though the backward sprint training could enhance certain aspects of muscle power and agility, it may not support peak performance in repeated sprint tests as effectively as forward sprinting, particularly when maximum effort is required during each sprint. The observed RSA gains are likely underpinned by metabolic adaptations such as increased phosphocreatine and glycogen resynthesis capacity, enhanced glycolytic enzyme activity, and improved buffering of intramuscular acidity; all of which contribute to better sprint recovery and repeated high-intensity performance [48].

### Limitations and Future Directions

This study has several limitations that should be considered when interpreting the results. First, the absence of an active control group limits the ability to fully isolate the effects of the RBST and RFST interventions, as no comparison was made with athletes involved only in regular basketball training. Second, the small number of participants represents an additional limitation, as a larger sample size would have increased the statistical power and improved the generalizability of the findings. Third, the RSA protocol should have been performed in the backward direction to better examine the specificity hypothesis. Fourth, although both the RFST and RBST groups had comparable overall training exposure, it would have been beneficial to monitor training load across the 8-week period using external indicators (e.g., total distance covered) and/or internal indicators (e.g., perceived exertion ratings, heart rate responses). Finally, despite instructing participants to exert maximal effort and perform each repetition as quickly as possible, we were unable to assess the metabolic cost or markers of training intensity, such as heart rate or power output, during the RBST and RFST programs. Inclusion of these measures would have helped to quantify the physiological demands of each intervention and better contextualize the adaptations observed. Additionally, we acknowledge that the predicted age of PHV is associated with an error of approximately ±6 months [49], particularly in the absence of direct assessment of secondary sexual characteristics or androgen status. These unmeasured factors could result in misclassification of biological maturity, which may influence individual training responses and the observed adaptations. Future research should examine how backward sprint training impacts performance over longer periods and across varying age groups, sports, and levels of competition. Additionally, it would be interesting to explore the combined effects of both training approaches to determine whether synergistic benefits occur. Moreover, incorporating direct measures of strength and neuromuscular function, such as muscle architecture, joint kinetics, or force production, would help clarify the mechanisms driving the observed performance adaptations.

## 5. Conclusions

This study provides novel evidence comparing the effects of RBST and RFST on multiple components of physical performance in youth male basketball players. Both training methods were effective in improving sprinting, jumping, CoD, agility, and RSA, with no adverse events reported. However, RBST elicited superior improvements in Y-agility and SLJ performance, while RFST demonstrated a small advantage in RSA_best_. These findings suggest that RBST may be a particularly valuable training strategy for enhancing agility and jumping ability, which are frequently required in basketball-specific movement scenarios. For coaches and practitioners, incorporating RBST into in-season training, either as a replacement for low-intensity drill time or as a targeted physical development tool, may offer a time-efficient and sport-relevant method for enhancing multidirectional performance. Importantly, the simplicity of the protocol and lack of equipment requirements make RBST easily implementable across a range of competitive levels. Given that both RBST and RFST produced improvements in RSA, practitioners may select the direction of sprinting based on the specific movement demands of their athletes or the phase of the season.

## Figures and Tables

**Figure 1 sports-14-00016-f001:**
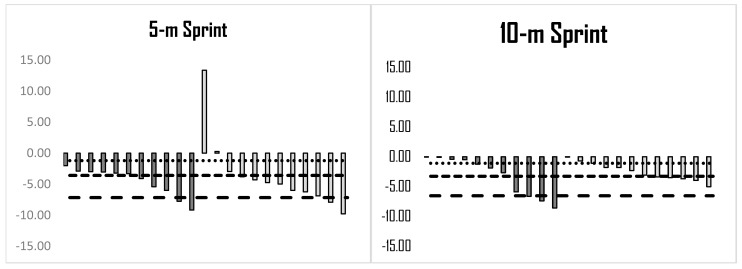
Individual percentage change from pre-training to post-training for performance tests relative to small, moderate, and large worthwhile changes in performance. 

 SWC; 

 MWC; 

 LWC; 
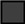
 RFST group; 
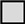
 RBST group.

**Table 1 sports-14-00016-t001:** Anthropometric characteristics of the included participants.

	RFST Group (*n* = 11)	RBST Group (*n* = 12)	*p* Value	t de Student (CI)
Age (years)	15.36 ± 0.63	15.20 ± 0.54	0.513	0.66 (−0.34 to 0.67)
Body height (cm)	177.36 ± 13.25	177.36 ± 11.02	0.992	−0.01 (−10.58 to 10.47)
Body mass (kg)	65.73 ± 13.98	62.42 ± 8.03	0.489	−0.70 (−6.46 to 13.08)
Maturity offset (years) *	1.86 ± 1.01	1.74 ± 0.68	0.743	0.33 (−0.62 to 0.85)
APHV	13.50 ± 0.61	13.46 ± 0.64	0.879	0.15 (−0.49 to 0.57)

Notes: Data are presented as means and standard deviations; RFST = repeated forward sprint training; RBST = repeated backward sprint training; *: as years from peak height velocity; APHV = age at peak height velocity; *n* = subjects. CI: confidence interval.

**Table 2 sports-14-00016-t002:** Repeated forward running training program vs. repeated backward running training program.

	RFST Sets × Reps × Distance (m) (Per Session)	RBST Sets × Reps × Distance (m) (Per Session)
Week 1	3 × 7 × 20	3 × 7 × 20
Week 2	3 × 7 × 20	3 × 7 × 20
Week 3	3 × 7 × 20	3 × 7 × 20
Week 4	2 × 7 × 20	2 × 7 × 20
Week 5	4 × 7 × 20	4 × 7 × 20
Week 6	4 × 7 × 20	4 × 7 × 20
Week 7	4 × 7 × 20	4 × 7 × 20
Week 8	3 × 7 × 20	3 × 7 × 20

RFST = repeated forward sprint training; RBST = repeated backward sprint training.

**Table 3 sports-14-00016-t003:** Descriptive performance testing results for RBST and RFST groups, including within-group changes from pre-training to post-training and between-group differences in the mean changes.

	Group	Pre (μ ± SD)	Post (μ ± SD)	Post-Pre % Difference (95% CL)	Post-Pre Training Effect Size	Difference RBST—RFST (μ ± SE)	RBST—RFST Effect Size (95% CL)
5 m sprint (s)	RBST RFST	1.17 ± 0.07 1.20 ± 0.07	1.10 ± 0.07 * 1.15 ± 0.06	−5.32 (−9.50 to −1.14) −4.58 (−5.92 to −3.23)	1.05 0.80	−0.01 ± 0.03	−0.10 (−0.92 to 0.72)
10 m sprint (s)	RBST RFST	1.93 ± 0.08 2.03 ± 0.11	1.86 ± 0.07 ^◊^ 1.96 ± 0.12 ^◊^	−3.53 (−4.89 to −2.17) −3.24 (−5.16 to −1.31)	0.98 −0.64	0.00 ± 0.02	−0.02 (−0.84 to 0.79)
20 m sprint (s)	RBST RFST	3.35 ± 0.16 3.49 ± 0.28	3.22 ± 0.14 ^◊^ 3.38 ± 0.23	−3.98 (−6.02 to −1.94) −3.27 (−5.78 to −0.77)	0.89 0.50	−0.02 ± 0.06	−0.10 (−0.92 to 0.72)
505 CoD (s)	RBST RFST	2.60 ± 0.32 2.78 ± 0.36	2.43 ± 0.25 * 2.72 ± 0.35	−6.48 (−11.46 to −1.50) −2.18 (−3.16 to −1.20)	0.62 0.18	−0.11 ± 0.96	−0.32 (−1.14 to 0.51) ^B^
Y-Agility (s)	RBST RFST	1.83 ± 0.22 1.63 ± 0.23	1.58 ± 0.17 ^†^ 1.62 ± 0.22	−13.27 (−18.03 to −8.52) −0.68 (−1.91 to 0.55)	1.33 0.05	−0.23 ± 0.05 ^◊^	−1.03 (−1.90 to −0.16) ^B^
CMJ (m)	RBST RFST	0.33 ± 0.05 0.28 ± 0.05	0.35 ± 0.06 * 0.30 ± 0.06	6.11 (1.81 to 10.39) 6.25 (2.24 to 10.39)	−0.34 −0.39	0.00 ± 0.08	0.05 (−0.77 to 0.87)
SLJ (cm)	RBST RFST	2.02 ± 0.17 1.83 ± 0.21	2.07 ± 0.18 ^†^ 1.86 ± 0.22 *	2.69 (1.72 to 3.65) 1.34 (2.04 to 1.63)	−0.12 −0.32	0.03 ± 0.08 *	0.16 (−0.66 to 0.98) ^B^
RSA_best_ (s)	RBST RFST	7.70 ± 0.33 8.15 ± 0.55	7.57 ± 0.29 7.92 ± 0.53 *	−1.74 (−3.70 to 0.22) −2.91 (−4.72 to −1.09)	0.44 0.45	0.10 ± 0.11	0.23 (−0.59 to 1.05) ^F^
RSA_mean_ (s)	RBST RFST	7.99 ± 0.32 8.48 ± 0.52	7.82 ± 0.25 * 8.26 ± 0.53 ^◊^	−2.16 (−3.92 to −0.39) −2.58 (−3.43 to −1.73)	0.62 0.43	0.05 ± 0.08	0.11 (−0.71 to 0.93)

RBST = repeated backward sprint training group; RFST = repeated forward sprint training group; CL = confidence limit; μ = mean; SD = standard deviation; SE = standard error; CoD = change of direction; CMJ = countermovement jump; SLJ = standing long jump; RSA_best_ = repeated sprint ability for fastest time; RSA_mean_ = mean repeated sprint ability time; ^B^ = training effect towards RBST; ^F^ = training effect towards RFST; * = *p* ≤ 0.05; ^◊^ = *p* ≤ 0.01; ^†^ = *p* ≤ 0.001.

## Data Availability

The raw data supporting the conclusions of this article will be made available by the authors upon request.

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
