# Peer review of "Effects of Repeated Forward Versus Repeated Backward Sprint Training on Physical Fitness Measures in Youth Male Basketball Players"

_sports, 2026, doi:10.3390/sports14010016_

Round 1

Reviewer 1 Report

Comments and Suggestions for Authors

Maturation status & equation

Please resolve the “post-pubertal” versus “pubertal” wording by explicitly defining the maturation classification rule (e.g., post-PHV defined as maturity-offset > 0 years) and ensure this terminology is used consistently in the Abstract, Methods, Results, and Tables.

In Methods, report the exact maturity-offset equation and its source (e.g., Mirwald et al., 2002, or the sex-specific Moore et al., 2015 update), including all variables and units required.

Describe the matching procedure you used for maturation status prior to randomisation (criteria and thresholds) and present baseline maturity-offset values by group with the corresponding between-group p-value/CI to document balance.

In Limitations, acknowledge potential residual error in maturity-offset methods, the absence of secondary sexual characteristics and (if applicable) unmeasured menstrual/androgen status, and discuss how any misclassification could influence training responses.

Statistical transparency

Report exact p-values and 95% CIs for all tests (main effects, interactions, pairwise contrasts), not only thresholds; provide standardised effect sizes with CIs (you already show ES CIs in Table 3 lines 205–211/263–276; keep and ensure consistency with p-values).

Author Response

Responses to Reviewer 1 were highlighted in yellow

Comments and Suggestions for Authors

Maturation status & equation

Comment

Please resolve the “post-pubertal” versus “pubertal” wording by explicitly defining the maturation classification rule (e.g., post-PHV defined as maturity-offset > 0 years) and ensure this terminology is used consistently in the Abstract, Methods, Results, and Tables.

Authors’ reply

We thank the reviewer for highlighting the need for clarity and consistency regarding the maturation classification terminology. In the revised manuscript, we have explicitly defined the maturation classification rule as follows: Based on the predicted maturity offset, athletes were classified according to their proximity to peak height velocity (PHV). Specifically, participants were categorized into three maturity groups: pre-PHV (–3 to –1 years from PHV), circa-PHV (–1 to +1 years from PHV), and post-PHV (+1 to +3 years from PHV).

Comment

In Methods, report the exact maturity-offset equation and its source (e.g., Mirwald et al., 2002, or the sex-specific Moore et al., 2015 update), including all variables and units required.

Authors’ reply

Thank you for your comment. The equation used was the one validated by Moore et al. (2015) The following statement was included in the main text: “The maturity offset method was used to assess the biological maturity of participants [26]. The following prediction equation was applied: Maturity offset = -7.999994 + (0.0036124*age*height).”

Moore SA, McKay HA, Macdonald H, Nettlefold L, Baxter-Jones AD, Cameron N, et al. Enhancing a Somatic Maturity Prediction Model. Med Sci Sports Exerc 2015; 47:1755–64.

Comment

Describe the matching procedure you used for maturation status prior to randomization (criteria and thresholds) and present baseline maturity-offset values by group with the corresponding between-group p-value/CI to document balance.

Authors’ reply

Thank you for this valuable comment. We have now clarified the matching procedure and provided explicit details on the criteria used. Specifically, participants were classified according to their maturity-offset–based pubertal status (pre-PHV, circa-PHV, or post-PHV). Prior to randomization, players were matched based on this biological maturation classification to ensure comparable maturity distributions between groups. All postpubertal players were evenly distributed between the RFST and RBST groups, and the four pubertal players were also balanced (two in each group). We have also presented the baseline maturity-offset values with the associated between-group p-value and confidence interval. These revisions have been incorporated into the manuscript (Methods section), and the text now reads:

“All participants demonstrated positive maturity offset values, confirming that the vast majority were postpubertal. Specifically, only two players in the RBST group and two players in the FRST group were classified as pubertal. Prior to randomization, players were matched according to their maturity offset to ensure similar biological maturation between groups. Before randomization, participants were matched based on their biological maturation status. All postpubertal players were randomly distributed evenly across the two groups, and the four pubertal players were also balanced, with two randomly assigned to the FRST group and two to the RBST group. The mean maturity offset values did not differ significantly between the FRST and RBST groups at baseline (FRST: 1.86 ± 1.01 years; RBST: 1.74 ± 0.68 years; p = 0.78, 95% CI = -0.28 to 0.22), indicating balanced maturation status distribution between the groups.”

  Comment

In Limitations, acknowledge potential residual error in maturity-offset methods, the absence of secondary sexual characteristics and (if applicable) unmeasured menstrual/androgen status, and discuss how any misclassification could influence training responses.

 Authors’ reply

We thank the reviewer for this important comment. We acknowledge that the estimation of maturity offset may carry some residual error (±6 months) (Mirwald et al. 2002), particularly in the absence of direct assessment of secondary sexual characteristics or androgen status. These unmeasured factors could result in misclassification of biological maturity, which may influence individual training responses and the observed adaptations. We have added a statement in the Limitations section to highlight that these factors should be considered when interpreting the results, and that future studies could benefit from more precise assessment of biological and hormonal maturity.

The following statement was added to the limitations: “Additionally, we acknowledge that the predicted aged of PHV is associated with an error of approximately ±6 months [53], particularly in the absence of direct assessment of secondary sexual characteristics or androgen status. These unmeasured factors could result in misclassification of biological maturity, which may influence individual training responses and the observed adaptations.”

Statistical transparency

Comment

Report exact p-values and 95% CIs for all tests (main effects, interactions, pairwise contrasts), not only thresholds; provide standardised effect sizes with CIs (you already show ES CIs in Table 3 lines 205–211/263–276; keep and ensure consistency with p-values).

Authors’ reply

We thank the reviewer for the suggestion. Exact p values, and confidence intervals have now been added for all effect size values and are reported consistently across the Results section

Reviewer 2 Report

Comments and Suggestions for Authors

Dear authors,

I recommend a major revision (41 comments).

Please review the PDF document where the reviewer has clearly indicated the details that need to be corrected in order for the manuscript to reach a high-quality standard.

Author Response

Responses to reviewer 2 were highlighted in green

Comment 1

Change of direction (COD) OR change of direction speed (CODS)

Authors’ reply

Changed as suggested throughout the whole manuscript

Comment 2

RSA

Authors’ reply

Changed as suggested.

Comment 3

This term is unfamiliar in sports practice. Please use change of direction (COD) throughout the text.

Authors’ reply

Changed throughout the whole article

Comment 4

This is correct. To highlight the importance of these abilities, please provide more specific information about how many jumps, sprints, and COD basketball players perform, as well as the duration of these activities.

Authors’ reply

Thank you for your suggestion. More details were added as suggested

“Basketball is characterized by its highly intermittent nature, with players performing approximately 1,050 distinct movements during a single competitive match equating to a change in speed or direction roughly every two seconds [9]. During gameplay, elite basketball athletes typically cover about 991 meters through high-intensity efforts, executing on average 40–60 maximal jumps and 50–60 rapid accelerations or changes of direction (CODs) [9]. Moreover, it has been reported that nearly half (≈52%) of all sprints in basketball involve at least one fast COD, emphasizing that the ability to efficiently execute directional changes, both with and without ball possession, is a fundamental determinant of performance [10].”

Comment 5

In the introduction, you repeat the same information three times at the beginning of the first, second, and third paragraphs

Authors’ reply

We revised the introduction as suggested. Thank you

Comment 6

The reviewer said that we have already mentioned the following sentence in the discussion:

The multi-directional nature of basketball requires athletes to use a variety of locomotion strategies, including BR, during both offensive and defensive manoeuvres [25,26]. BR has gained attention as an effective training stimulus for enhancing various physical attributes, including sprinting, jumping, CoD speed, and RSA [23,24, 27,28,29].

Authors’ reply

The authors agree with the reviewer and we removed the sentence as suggested. Thank you

Comment 7

Try to emphasize in this sentence that this is the first study that compares.

Authors’ reply

Added as suggested. Thank you. The revised statement reads: “To the best of the authors’ knowledge, the current study is the first to compare the effects of RBST versus RFST on various measures of physical fitness in youth male basketball players.”

Comment 8

There is no need to cite authors when stating the hypothesis

Authors’ reply

Removed as suggested. Thank you

Comment 9

The reviewer suggests to change materials to Materials and Methods

Authors’ reply

Added as suggested

Comment 10

This information should be included within the Training Program subsection.

Authors’ reply

We added this information to the training program as suggested. The revision now reads: “

Each training session lasted approximately 80–90 minutes in total, consisting of a 15-20-minute standardized warm-up (mobility, dynamic stretching, and activation drills), 10–15 minutes of the RFST or RBST intervention, and approximately 40–45 minutes of technical–tactical basketball drills and scrimmage play. Following the RFST and RBST sessions, players continued their regular basketball training, which included offensive and defensive drills as well as small-sided games. At the end of each session, a 5–10-minute cool-down with low-intensity jogging and stretching was performed”

Comment 11

Please provide some additional information about the measurement procedures and the organization of the testing.

Authors’ reply

More information was added as suggested: 

“Testing occurred over three days with anthropometric measurements, linear sprint speed, CoD speed, and agility testing conducted on the first day, jump testing on the second day, and RSA on the third day.”

Comment 12

This should be the first subsection within the Materials and Methods section.

Authors’ reply

Done as suggested

Comment 13

You did not provide enough details about the training program. How long did the first, second, and third parts last, and what was the total duration of the entire training session? What was implemented at the beginning and at the end of the training? Did you monitor training intensity, and if so, how?

Authors’ reply

Thank you for your legitimate and pertinent comment. Accordingly, more details were added in the training program section as follows:

“Each training session lasted approximately 80–90 minutes in total, consisting of a 15-20-minute standardized warm-up (mobility, dynamic stretching, and activation drills), 10–15 minutes of the RFST or RBST intervention, and approximately 40–45 minutes of technical–tactical basketball drills and scrimmage play. Following the RFST and RBST sessions, players continued their regular basketball training, which included offensive and defensive drills as well as small-sided games. At the end of each session, a 5–10-minute cool-down with low-intensity jogging and stretching was performed.

Please note that there was no direct monitoring of training load using external or internal training load tools. This is acknowledged as a limitation of the study. However, training exposure in terms of duration and frequency was comparable across both groups, suggesting that overall training load was similar. Nevertheless, as indicated, the use of internal or external training load monitoring tools would have been preferable. 

“Fourth, although both the RFST and RBST groups had comparable overall training exposure, it would have been beneficial to monitor training load across the 8-week period using external indicators (e.g., total distance covered) and/or internal indicators (e.g., perceived exertion ratings, heart rate responses).”

Comment 14

This should not be a separate section, but rather included within the subsection Materials and Methods.

Authors’ reply

Relocated as suggested. Thank you

Comment 15

These 2 sentences should be included within the subsection Experimental approach to the problem

Authors’ reply

Done as suggested. Thank you

Comment 16

Please confirm the hypothesis. Are the results in line with expectations or not?

Authors’ reply

Thank you for your comment. Accordingly, the following revision was integrated: “These findings are generally consistent with our initial hypothesis, which proposed that both repeated forward and backward sprint training would enhance physical performance, with greater improvements expected following backward sprint training.”

Comment 17

The reviewer asks to remove the following sentence:

Meanwhile, research examining RFST in youth soccer players has reported small-to-moderate (ES = 0.32–0.68) improvements in 5-m to 30-m linear sprint performance after six to nine weeks of training 327 with and without change of direction [4, 33].

Authors’ reply

The sentence was removed as suggested.

Comment 18

The reviewer asks to remove the following sentence: Change of direction, speed, and agility are fundamental physical capacities in basketball, underpinning rapid positional adjustments, defensive reactions, and multidirectional movements in response to unpredictable stimuli [36, 39, 41].

Authors’ reply

Removed as suggested. Thank you

Comment 19: Let me remind on again to use the abbreviation CoD

Authors’ reply

Changed throughout the whole article.

Comment 20

Previous research should be previous studies

Authors’ reply

Changed as suggested.

Comment 21.

You are comparing your results too much with these two studies. Is there any similar study where this topic has been investigated in other team sports, for example in handball or volleyball?

Authors’ reply

Other references were added

Sammoud S, Bouguezzi R, Uthoff A, Ramirez-Campillo R, Moran J, Negra Y, Hachana Y, Chaabene H. The effects of backward vs. forward running training on measures of physical fitness in young female handball players [Original Research]. Frontiers in Sports and Active Living. 2023; 5.

Negra Y, Sammoud S, Uthoff A, Ramirez-Campillo R, Moran J, Chaabene H. The effects of repeated backward running training on measures of physical fitness in youth male soccer players. J Sports Sci. 2023; 40(24): 2688-2696.

Comment 22

I don’t know why you are using a period after the citation of references. That’ s a mistake

Authors’ reply

You’re right. Removed. Thanks

Comment 23

Reference?

Authors’ reply

Added as suggested.

Comment 24

Limitations and Future Directions After the limitations, present the recommendations for future researchers.

Authors’ reply

Corrected as suggested.

Comment 25

The second limitation is the small number of participants.

Authors’ reply

Added as suggested.

Comment 26

Replace this term. 'Horizontal power' is not a common term in our field. CMJ and SLJ explosive power, and both are jumps performed in the sagittal plane, with one measuring jump height and the other measuring jump distance.

Authors’ reply

Corrected and replaced by jumping ability

Comment 27

Replace this term. 'Horizontal power' is not a common term in our field. CMJ and SLJ explosive power, and both are jumps performed in the sagittal plane, with one measuring jump height and the other measuring jump distance.

Authors’ reply

corrected and replaced by jumping ability

Comment 28

This text should be included within the subsection Limitations and Future Directions.

Authors’ reply

Done as suggested.

Comment 29

Pay attention to the MDPI reference citation format.

Authors’ reply

All references were double-checked and corrected as suggested.

Round 2

Reviewer 1 Report

Comments and Suggestions for Authors

There is a logic conflict: Table/Methods say two players per group were pubertal, yet in Results they write “The maturation level of all participants was classified as postpubertal.” Both cannot be true; please harmonize with the actual offset values in Table 1.

Table 3 still shows implausible dispersion for CMJ at baseline (RBST 33.16 ± 35.18 cm) and a post value that flips to a plausible SD (35.18 ± 5.94 cm), strongly suggesting a formatting/entry error. The SLJ “Difference (RBST−RFST)” line remains odd (2.96 ± 0.08 with a trivial ES that has a CI crossing zero); please re-check calculations and units. The table footnote again calls RFST the “control group,” which is inaccurate because both arms trained. Abbreviation usage is better but still mixed in places (FRST vs RFST; RSAavg vs RSAmean).

Author Response

Responses to Reviewer 1 were highlighted in yellow

Comments and Suggestions for Authors

Comment1

There is a logic conflict: Table/Methods say two players per group were pubertal, yet in Results they write “The maturation level of all participants was classified as postpubertal.” Both cannot be true; please harmonize with the actual offset values in Table 1.

Authors’ reply:

We thank the reviewer for pointing out this inconsistency and apologize for it. We have corrected the Results section to accurately reflect the maturity offset values reported in Table 1.

Based on the maturity offset values (Table 1), the majority of participants were classified as postpubertal; however, four players (two in each group) fell within the pubertal range. Accordingly, maturation status was not uniform across all participants, but the distribution of pubertal and postpubertal athletes was balanced between groups”

Comment 2

Table 3 still shows implausible dispersion for CMJ at baseline (RBST 33.16 ± 35.18 cm) and a post value that flips to a plausible SD (35.18 ± 5.94 cm), strongly suggesting a formatting/entry error.

Authors’ reply

Thank you for bringing this typo to our attention. This was corrected.

Comment 3

The SLJ “Difference (RBST−RFST)” line remains odd (2.96 ± 0.08 with a trivial ES that has a CI crossing zero); please re-check calculations and units.

Authors’ reply:

We appreciate the reviewer’s careful examination of the SLJ “Difference (RBST−RFST)” values. We have re-checked all calculations, units, and the original dataset to ensure accuracy. The difference value of 2.96 ± 0.08 cm, along with the corresponding effect size and confidence interval, was verified and confirmed to be correct.

Comment4

The table footnote again calls RFST the “control group,” which is inaccurate because both arms trained. Abbreviation usage is better but still mixed in places (FRST vs RFST; RSAavg vs RSAmean).

Authors’ reply

Double checked and corrected throughout the whole article

Reviewer 2 Report

Comments and Suggestions for Authors

The authors have thoroughly addressed all previously provided reviewer comments. Appropriate revisions and clarifications have been implemented throughout the manuscript, which have notably improved the overall quality, clarity of presentation, and scientific contribution of the paper.

The revised version of the manuscript demonstrates that the authors carefully considered the suggestions and successfully enhanced key methodological and interpretative aspects. Based on the current form and the improvements made, I believe that the manuscript now meets the journal’s standards and is suitable for publication without further modifications. 

Author Response

Responses to the reviewer 2 were highlighted in green

Comments and Suggestions for Authors

The authors have thoroughly addressed all previously provided reviewer comments. Appropriate revisions and clarifications have been implemented throughout the manuscript, which have notably improved the overall quality, clarity of presentation, and scientific contribution of the paper.

The revised version of the manuscript demonstrates that the authors carefully considered the suggestions and successfully enhanced key methodological and interpretative aspects. Based on the current form and the improvements made, I believe that the manuscript now meets the journal’s standards and is suitable for publication without further modifications. 

Authors’ reply

Thank you for your positive evaluation and constructive feedback during the first round. We very much appreciate your time.